# Chilling Tolerance in Maize: Insights into Advances—Toward Physio-Biochemical Responses’ and QTL/Genes’ Identification

**DOI:** 10.3390/plants11162082

**Published:** 2022-08-09

**Authors:** Yun Ma, Renxiang Tan, Jiuran Zhao

**Affiliations:** 1Institute of Maize, Beijing Academy of Agriculture and Forestry Sciences, Beijing 100097, China; 2College of Plant Science and Technology, Beijing University of Agriculture, Beijing 100096, China

**Keywords:** cold tolerance, maize, low temperature, membrane system, antioxidants, photosynthesis, osmoregulatory substances, plant hormones

## Abstract

Maize, a major staple cereal crop in global food supply, is a thermophilic and short-day C4 plant sensitive to low-temperature stress. A low temperature is among the most severe agro-meteorological hazards in maize-growing areas. This review covers the latest research and progress in the field of chilling tolerance in maize in the last 40 years. It mainly focuses on how low-temperature stress affects the maize membrane and antioxidant systems, photosynthetic physiology, osmoregulatory substances and hormone levels. In addition, the research progress in identifying cold-tolerance QTLs (quantitative trait loci) and genes to genetically improve maize chilling toleranceis comprehensively discussed. Based on previous research, this reviewprovides anoutlook on potential future research directions and offers a reference for researchers in the maize cold-tolerance-related field.

## 1. Introduction

Maize, an important staple food crop, is widely planted worldwide, with its planting area and production third-greatest of all crops, surpassed only by rice and wheat. It is also used as a feed crop, energy crop, and source ofindustrial raw materials. Thus, maize has a wide range of uses and applications, including food, feed for fishery and livestock farming, biogas generation, paper production, industrial raw materials (ethanol production) and food processing (edible oil and starch). Furthermore, fermentation science advances have led to maize’s utilization for renewable, inexpensive and environmentally friendly biofuels [1].

Maize is a thermophilic crop originating from Mexico with an optimum growth temperature of 25–28 °C and a minimum growth temperature of 6–10 °C. It is susceptible to temperature stress below 12 °C [2]. Low-temperature stress has multiple adverse effects on the maize morphology and physiology. Morphologically, low-temperature stress reduces seed emergence and seedling establishment, leading to leaf wilting and chlorosis; severe cold damage can result in rotted seeds, necrosed leaves and even plant death [3,4]. A low temperature affects the root system architecture, reducing root biomass, surface area and volume. It restricts the acquisition of water and nutrients, ultimately hampering the shoot growth under chilling stress [5,6] (Figure 1). In terms of physiology, cold stress causes various changes in membrane fluidity, the antioxidative defense system and photosynthetic performance, among others [7,8,9]. Although cultivation of maize has expanded into higher latitudes, low-temperature stress injury is still one of the most severe agro-meteorological hazards faced in temperate, tropical and subtropical maize-growing regions during sowing. Fluctuating weather, especially low temperatures, always causes risks of yield loss [2,3,10]. Therefore, it is crucial to fully elucidate maize cold-tolerance mechanisms to improve its robustness against cold injuries. This can aid maize in maintaining its growth under low-temperature stress conditions and minimizepotential yield losses. In the last few decades, considerable efforts have been undertaken to understand the mechanism underlying the physio-biochemical responses to low-temperature stress and determine QTLs/genes controlling chilling tolerance.

Under normal conditions (left panel), the plants have healthy green leaves and developed roots. In contrast, chilling stress (right panel) causes symptoms such as slow growth, dwarf plants, wilted and chlorotic leaves and reduced length, surface area and root volume.

As presented here, a literature search from the National Center of Biotechnology Information (NCBI) was conducted to review the literature published in the field of maize cold tolerance in the last 40 years (1982–2022). This review summarizes the research progress in this field and provides an outlook and future research directions toward improving maize cold tolerance.

## 2. Effect of Low Temperature

Low temperatures can affect vital cell functions and eventually plant growth and development. Low temperatures are divided into chilling and freezing according to the temperature range. The former is lower than20 °C and the latter is below 0 °C. Low-temperature stress encountered by maize usually refers to chilling. Maize is sensitive to low-temperature stress and slows its growth below 12 °C, then growth is eventually halted when the temperature drops to approximately 6 °C. Furthermore, lower temperatures may lead to irreversible damage to maize tissues and cells [2]. Among the different growth and developmental stages of maize, a low temperature most significantly affects the germination, seedling and flowering stages. Physiologically, low-temperature stress affects the seed germination process, seedling vigor, overall germination rate and potential and root system architecture. Therefore, at the seedling stage that follows, low-temperature stress affects the degree of germination, seedling rate, and seedling potential, resulting in plant dwarfism and significantly impacting the stem and root development. A low temperature at the reproductive stage affects ear differentiation at the tasseling stage, pollination at the flowering stage, the grain-filling rate, dry matter accumulation and grain maturity. At the molecular cytological level, the effects of a low temperature on maize primarily manifest with changes in the membrane structure, photosynthetic physiology, osmoregulatory and antioxidant enzymes and hormone levels. These changes are some of the most investigated aspects in maize chilling tolerance-related research and are addressed individually in this review.

### 2.1. Effect of Low-Temperature Stress on the Membrane System

Low temperatures lead to changes in the structure and shape of maize cell membranes, enhance the cell membrane permeability and cytoplasmic extravasation and decrease hydraulic conductivity. More importantly, damage to cell membranes in cold-sensitive maize genotypes under low-temperature stress prevents stress recovery due to impaired root hydraulic conductivity and loss of cellular integrity [11]. The sensitivity of seeds to a low temperature during imbibition depends on the accumulation of saturated or low unsaturated fatty acids. A lower electrolyte leakage rate during low-temperature imbibition in atolerant hybrid was correlated with the accumulation of polyunsaturated chains [12]. Using a B73 (sensitive variety) and IABO78 (tolerant variety) F_2_ population with differences in germination activity under low-temperature conditions, it was observed that the F_2_ lines with high germination rates, under a low temperature (9.5 °C), showed higher mobility and a higher proportion of C18 unsaturated fatty acids in the mitochondrial inner membrane compared to F_2_ lines with low germination rates [13]. In addition, chilling-tolerant inbred lines had higher water potential, osmotic potential, and diffusion resistance values than chilling-sensitive lines [14]. Lipidomics and transcriptomics analyses revealed an increase in phosphatidic acid and digalactosyl diacylglycerol contents, a decrease in phosphatidylcholine and monogalactosyl diacylglycerol contents, changes in lipid synthesis and active regulation of membrane lipids under low-temperature conditions [15]. Furthermore, compared to chilling-sensitive inbred lines, tolerant maize lines accumulated more unsaturated fatty acids and secondary metabolites to maintain the stability of cell membranes and protect their photosystem structure [16]. In the senescing chloroplasts of low-temperature-sensitive maize inbred lines, oxygen evolution inactivation and free fatty acid accumulation occurred; however, in the tolerant lines, oxygen release decreased obviously, without an increase in free fatty acids’ accumulation and leaf senescence [17]. In addition, the cell wall biochemical components closely related to the cell membrane structure were also altered at low temperatures [18].

### 2.2. Effect of Low-Temperature Stress on the Antioxidant System

A low temperature increases oxidative stress and the intracellular H_2_O_2_ (hydrogen peroxide), O_2_^−^ (superoxide anion), hydroxyl ion and malondialdehyde contents [5,11,19]. To counteract oxidative stress, plants respond by increasing total glutathione, glutathione reductase activity [20] and catalase and esterase expression levels [21]. Low temperature-induced oxidative stress causes leakage of intracellular organic acids, enzyme inactivation and protein and DNA damage. It promotes fungal and bacterial growth, leading to seeds’ decay, powdering and rotting [22]. To alleviate low-temperature stress-induced oxidative damage, seed priming with melatonin or chitosan has been shown to increase the activity of antioxidant enzymes and effectively improve the germination potential and grate at low temperatures [22,23]. Seeds coated with salicylic acid exhibit enhanced activity of protective enzymes, a higher germination rate and longer root and stem lengths under low-temperature stress [24]. Adding silica to the plant growth medium increases the activity of superoxide dismutase and the antioxidant content [25]. Exogenous administration of diethyl aminoethyl hexanoate inhibits the low-temperature-mediated generation of O_2_^−^ and the increase in H_2_O_2_ content while maintaining the net photosynthetic rate, thus enhancing maize seedlings’ cold tolerance [26]. Leaves of the low-temperature-tolerant inbred line CO 328 exhibited higher superoxide dismutase activity than the sensitive line CO 316 [8]. Furthermore, cold-acclimated seedlings had higher catalase, glutathione reductase and guaiacol peroxidase activities during the low-temperature treatment and recovery stages compared to non-acclimated seedlings [27]. In addition, pretreatment of 3-day-old maize seedlings with H_2_O_2_ or menadione at room temperature induced cold tolerance via increased catalase3 and guaiacol peroxidase activities [19]. Higher superoxide dismutase, peroxidase, catalase and glutathione reductase activities were observed in the hybrid XD889, which had a longer root length, higher root biomass and larger root volume and surface area under low-temperature stress compared to the hybrid XD319 and the inbred lines Yu13 and Yu37 [5]. *ZmASR1* (ABA-stress-ripening) overexpression in maize reduced the malondialdehyde content, increased superoxide dismutase and peroxidase activities and increased cold tolerance [28]. The overexpression of the R2R3-MYB transcription factor *ZmMYB31* of maize in *A*. *thaliana* effectively suppressed ROS (reactive oxygen species) accumulation caused by a low temperature. Overexpression of *ZmMYB31* in *A*. *thaliana* also enhanced the superoxide dismutase and ascorbate peroxidase activities, thereby enhancing cold tolerance in transgenic *A*. *thaliana* plants [29]. *ZmMKK1* overexpression in tobacco resulted in greater antioxidant enzyme activity, accumulation of osmoregulatory substances and upregulated expression of ROS and stress response-related genes, thereby increasing transgenic tobacco cold tolerance [30].

### 2.3. Effect of Low-Temperature Stress on Photosynthetic Physiology

A low temperature significantly affects photosynthesis in maize plants, resulting in reduced maximum catalytic activity of photosynthetic enzymes [31], lowered photosystem II activity [18,32,33] and leaf chlorophyll content, CO_2_ assimilation [34] and reduced protein content [31]. Several cold acclimation and light intensity experiments revealed that exposure to high-intensity light during cold acclimation enhanced subsequent adaption to severe cold stress. It also increased the low-temperature tolerance and soluble sugar content [35]. The chloroplast photosynthetic efficiency and ultrastructure were more significantly affected in the low-temperature-sensitive inbred line CM 109 compared to the tolerant inbred line KW 1074 [36]. Photosynthesis reduction in maize under low-temperature conditions was associated with plasmodesmata closure between mesophyll cells and bundle sheath cells and between bundle sheath cells and vascular parenchymal cells [37]. Maize seeds exposed to low-temperature stress during germination maintained their growth by regulating photosynthesis, with more severe damage to the photosynthetic system observed in chilling-sensitive maize genotypes [38]. After cold acclimation (14/12 °C), chilling-tolerant maize exhibited lower photosynthetic apparatus damage by cold stress (8/6 °C) [39]. Overexpression of *RAF1*–*LSSS* (*Rubisco assembly factor 1*-*Rubisco large and small subunits*) in maize improved photochemical quenching before and after low-temperature treatment and accelerated maize recovery [40]. Overexpression of the R2R3–MYB transcription factor *ZmMYB*–*IF35* in *A*. *thaliana* alleviated photosystem II photoinhibition and increased antioxidant enzyme activity, thereby enhancing low-temperature tolerance [1].

### 2.4. Effect of Low-Temperature Stress on Osmoregulatory Substances

Osmoregulatory substances such as soluble sugars and proline increase significantly under low-temperature stress to prevent cellular water loss and maintain normal physiological activities [23,30]. After low-temperature treatment, sucrose rapidly accumulates in sensitive maize genotypes, causing changes in the cellular osmotic potential and decreasing the cell wall pectin content and pectin esterase activity [41]. In cold-tolerant maize genotypes, there is a significant increase in cryoprotectants such as free proline and soluble sugars. At the same time, stress markers (e.g., malondialdehyde) are maintained at low levels, thus contributing to a robust antioxidant defense system [42]. Chilling-tolerant genotypes had greater glucose-6-phosphate dehydrogenase activity and a higher sucrose-to-starch ratio than sensitive genotypes, implying that starch–sucrose metabolism is a principal determinant of the low-temperature response [43]. Furthermore, *ZmDREB1A* increased raffinose biosynthesis by regulating *ZmRAF* (raffinose synthase) expression, a key enzyme in raffinose biosynthesis, thus improving low-temperature tolerance [44].

### 2.5. Effect of Low-Temperature Stress on Plant Hormone Levels

Plant hormone levels change under low-temperature conditions, with ABA and SA levels increasing and GA levels decreasing [45]. Exogenous hormone application affects maize cold tolerance. For example, ABA application before cold exposure induces changes in protein synthesis, causing maize suspension culture cells to acquire higher low-temperature tolerance [46]. Similarly, maize seedlings pretreated with ABA exhibit enhanced antioxidant enzyme activity and greater cold tolerance [47]. The ability of maize to survive after low-temperature stress is highly correlated with ABA synthesis upregulation, as evidenced by transcriptomic data from roots, leaves and crowns [48]. Spraying maize radicles or leaves with SA before low-temperature treatment reduces leaf electrolyte leakage and increases the activity of glutathione reductase and guaiacol peroxidase, thereby enhancing chilling tolerance [49,50]. The combined application of SA and H_2_O_2_ induces the expression of antioxidant enzymes, GA synthesis and GA and ABA signaling genes, as well as causing accelerated seed germination under low-temperature stress, improved seed vigor and elevated seedling growth [51]. EBR (24-epibrassinolide) treatment of maize seedlings results in higher tissue water content, lower membrane damage, increased chlorophyll, soluble sugar and protein contents and enhanced tolerance to a low temperature [52]. Furthermore, overexpression of *ZmPgb1.1* (*Phytoglobin*) in maize inhibits nitric oxide and BR (brassinolide) biosynthesis, as well as the BR response, thereby alleviating low-temperature stress [53].

## 3. Identification and Localization of Chilling Tolerance-Related QTL/Genes in Maize

Low-temperature stress can affect a variety of physiological and biochemical indicators, as mentioned above. Therefore, it is feasible to enhance maize cold tolerance by modifying the underlying genes controlling these traits. Accordingly, more research is needed to identify the genetic loci that control chilling tolerance in maize. Among the published literature on chilling tolerance in maize, only three articles used roots, protoplasts and suspension cells as experimental materials. Besides these, approximately 30 percent of papers used seeds as experimental materials. Most used 2–4-leaf-stage maize seedlings as experimental material for low-temperature treatment (Table 1 and Table 2). Yet, the two most vulnerable periods to low-temperature stress during field growth are seed germination and early seedling growth. Low-temperature treatments conducted during these growth stages provide insights into the time actual field-grown plants are most likely to be affected by low-temperature climates. Thus, they are effective at revealing the adverse effects of low-temperature stress during early maize growth on later growth and yields [52].

SSR, simple sequence repeat markers; SNP, single nucleotide polymorphism; RFLP, restriction length polymorphism fragments; RIL, recombinant inbred lines; Φ_PSII_, operating quantum efficiency of the photosystem II (PSII) photochemistry; Fv/Fm, maximal quantum efficiency of PSII primary photochemistry; SPAD, chlorophyll content; Fo, minimum fluorescence; Fm, maximum fluorescence; SDW, shoot dry weight; LAT, area of the third leaf; N%, nitrogen content; C%, carbon content; C:N, ratio of carbon-to-nitrogen content; LDW, dry weight of the second leaf; LAS, area of the second leaf; SLA, specific leaf area (calculated as the LAS:LDW ratio); RGR, relative growth rates; GP, germination percentage; GI, germination index; SL, seedling length; SVI, simple vigor index; MGT, mean germination time; GR, germination rate; DT50, days to 50% germination; GI, germination index; RGR, relative germination rate, calculated by dividing the germination rate under the chilling condition by the germination rate under the normal condition; RDT50, relative DT50, calculated by dividing the days to 50% germination under the chilling condition by the days to 50% germination under the normal condition; RGI, relative GI, calculated by dividing the germination index under the chilling condition by the germination index under the normal condition; growth ratio, calculated by dividing the average length of roots and shoots of each variety at a low temperature by the average length of roots and shoots of each variety at a control temperature; LRD, leaf roll degree; WCS, water content of shoots and leaves; RRS, ratio of root-to-shoot; SSC, soluble sugar content; RGR, relative germination rate; RGL, relative germ length; RRL, relative radicle length; RRSA, relative radicle surface area; RRV, relative radicle volume; RGI, relative germination index; RVI, relative vitality index; RSVI, relative simple vitality index; XYRGR, Xiang Yang relative germination rate; XYRGL, Xiang Yang relative germ length; XYRSVI, Xiang Yang relative simple vitality index; KSRGR, Ke Shan relative germination rate; KSRGL, Ke Shan relative germ length; KSRSVI, Ke Shan relative simple vitality index; LL, final leaf length; LA4, leaf area of the fourth leaf; LER, leaf elongation rate; CLma, mature cell length; CLme, meristematic cell length; *P*, cell production; *D*, division rate; *Tc*, cell cycle duration; *Lmer*, length of the meristem; *Lgr*, length of the growth zone; *Lel*, length of the elongation zone; *Nmer*, number of cells in the meristem; *Ngz*, number of cells in the growth zone; *Nel*, number of cells in the elongation zone; *Tel*, cell elongation duration; *Rel*, cell elongation rate; FG, germination rate (percentage of germinated seeds to the total seeds) at 5 days; TG, germination ratio at 10 days; RL, root length at 10 days; SL, shoot length at 10 days; RRS, ratio of root length to shoot length; DTE, days to emergence; DTSL, days to the second leaf; EV, early vigor; DW, dry weight of the aboveground biomass; LTPL, plumule length under low temperature; LTSL, seedling length under low temperature; LTRL, root length under low temperature; LTGR, germination rate under low temperature; LTGI, germination index under low temperature; LTVI, vigor index under low temperature; LTSVI, simple vigor index under low temperature; LTAGD, average germination days under low temperature.

### 3.1. Identification of Chilling Tolerance-Related QTLs/Genes during Germination

Using 1172 and 1139 SNP (single nucleotide polymorphism) markers in 208 and 212 recombinant inbred lines from the crosses of Yu82 × Shen137 and Yu537A × Shen137, respectively, genetic linkage analysis of indicators related to seed vigor traits at low temperatures was undertaken. Such indicators included the germination percentage, germination index, seedling length, single vigor index and mean germination time, and five possible mQTLs linked to improved seed vigor under low-temperature conditions were identified [56]. By comparing three germination indexes linked to low-temperature tolerance (germination rate, days to reach 50% germination rate and germination index) in 282 inbred lines at 25 and 8 °C, a genome-wide association analysis was performed. In this way, 18 candidate genes (associated with 17 significant SNPs) associated with cold tolerance during germination were identified [3]. Five QTL clusters formed by seven QTLs located on chromosomes 1, 2, 3, 4 and 9 were potentially associated with low-temperature germination-related traits (germination rate, germination index, mean germination days, root length, plumule length and seedling length) in a 176 IBM Syn10 doubled haploid population from the B73 × Mo17 cross. These five QTL clusters, containing 39 candidate genes, were compared for their expression levels and amino acid sequences using the RNA-seq results of the two parental lines (B73 and Mo17) before and after the low-temperature treatment. The differences in expression levels among *Zm00001d043166*, *Zm00001d007315* and *Zm00001d027974*, as well as the differences in amino acid sequences among *Zm00001d027976*, *Zm00001d007311* and *Zm00001d053703*, were identified as potentially responsible for the differences in low-temperature germination performance of B73 and Mo17 [65]. A genome-wide association analysis of 14 relative indices, such as the relative germination rate, relative germ length, relative radicle length, relative radicle surface area, relative radicle volume, relative germination index, relative vitality index, etc., in 222 maize inbred lines under low and normal temperatures, identified 82 candidate genes associated with 32 SNPs. According to the RNA-seq results of the low-temperature-tolerant inbred line Zao 8-3 and the sensitive inbred line Ji 853, among the 10 DEGs (differentially expressed genes) before and after low-temperature treatment, *Zm00001d039219*, involved in the MAPK signaling pathway, and *Zm00001d034319*, involved in the fatty acid metabolic process, were significantly upregulated in Zao 8-3 and downregulated in Ji 853, and thus may be key candidate genes underlying low-temperature tolerance [60]. Genome-wide association analysis of five germination-related traits (germination rate at 5 days, germination rate at 10 days, root length at 10 days, shoot length at 10 days and the ratio of root length to shoot length) in 300 inbred lines at a low temperature (10 °C) identified 15 significant SNPs, three of which were shared by multiple traits. Further candidate gene association analysis, haplotype analysis and expression, and functional analyses of homologous genes identified the low-temperature-tolerance germination-related candidate genes as *Zm00001d010458*, *Zm00001d050021*, *Zm00001d010454* and *Zm00001d010459* [62]. In another study, the varieties Picker and PR39B29 (cold-tolerant) and Fergus and Codisco (cold-sensitive) were used for gene expression analysis with a 46K microarray one day after germination. They differed in their growth ratios, calculated by dividing the average length of roots and shoots at a low temperature by the average length of roots and shoots at a control temperature. After low-temperature treatment, Fergus and Codisco showed no significant differences in gene expression levels. In contrast, a total of 64 DEGs were observed between Picker and PR39B29, four of which were shared. The expression levels of *MZ00004486*, encoding PRP-1 (pathogenesis-related protein 1), showed the same trend after low-temperature treatment between Picker and PR39B29, meaning it was considered a key gene involved in the low-temperature response [58].

### 3.2. Identification of Chilling Tolerance-Related QTLs/Genes during the Seedling Stage

Genome-wide association analysis of six indicators (days to emergence, days to second leaf, early vigor, relative leaf chlorophyll content in the second leaf, maximum quantum efficiency of photosystem II, dry weight of the aboveground biomass) in an 836 maize inbred line population subjected to normal and low-temperature conditions identified 159 QTLs containing 187 significant associated SNPs. However, most of the QTLs explained a low percentage of the phenotypic variation. They were specific to only certain environments, such as growth chamber trials under cold and control conditions and field trials of early sowing across three consecutive years. Furthermore, no QTL sexplaining a high percentage of variance were detected under low-temperature conditions [63]. Using photosynthetic indicators, two inbred lines differing in chilling tolerance were crossed, ETH–DH7 (chilling-tolerant) and ETH–DL3 (chilling-sensitive), and their F_2:3_ population was evaluated by sowing at two time points in two different years. The first time point of each year was designated as the low-temperature treatment group and the second as the normal-temperature control group. Several QTLs were identified as significantly associated with chlorophyll fluorescence parameters, leaf greenness, leaf area, shoot dry weight and shoot nitrogen content [54]. In another study, 375 maize inbred lines were subjected to low-temperature treatments in a growth chamber and the field. Subsequently, parameters related to photosynthesis and growth, including Φ_PSII_, Fv/Fm, relative chlorophyll content, shoot dry weight, leaf dry weight, area of the second leaf, specific leaf area and relative growth rates, were investigated. The QTLs identified corresponded to genes involved in ethylene signaling and BR and lignin biosynthesis, and were associated with chilling tolerance during the seedling stage [55]. In two populations consisting of 306 dent and 292 flint maize inbred lines, respectively, a genome-wide association analysis was performed on traits such as the number of days from sowing to emergence, relative leaf chlorophyll content, Fv/Fm and early vigor. Nine QTLs were identified containing multiple candidate genes significantly associated with the abovementioned traits, which can be implemented for cold-tolerance improvement [57]. Furthermore, a genome-wide association analysis and subpopulation association analysis were performed for four cold-tolerance traits (leaf roll degree, water content of shoots and leaves, ratio of root-to-shoot and soluble sugar content) in the seedling stage using 338 testcrosses, a cytoplasmic male sterile line S-Mo17 and 338 maize inbred lines as parents. In this way, 32 significant SNP loci were identified, linked to 94 candidate genes, 36 of which were associated with stress tolerance. Further semi-quantitative RT-PCR experiments suggested that 10 candidate genes were upregulated by a low temperature in hybrids using K932 (low-temperature sensitive) and Mei C (low-temperature tolerant) as parents. At the same time, only *GRMZM2G019986* was induced by a low temperature in the parent Mei C, suggesting that heterosis was involved in cold tolerance in maize. Further linkage analysis of cold-tolerance-related indicators during the seedling stage using the F_2:3_ population constructed by K932 and Mei C identified seven QTLs significantly associated with cold tolerance, six of which showed partial dominance or over-dominance [59]. Using a recombinant inbred line population (IBM Syn4 RIL) constructed from B73 and Mo17 (IBM) with significant differences in cold tolerance during seeding, a linkage analysis of chlorophyll concentration, leaf color and leaf damage under low-temperature stress revealed that a QTL of approximately 2.7 M in chromosome 5 of Mo17 and a QTL of approximately 2.3 M in chromosome 1 of B73 were associated with greater cold tolerance. Therefore, 27 candidate genes within these 2 QTLs responsive to low-temperature stress, with sequence differences between B73 and Mo17, are potentially associated with an early low-temperature response [64]. Samples from the meristematic, elongation and maturation zones of leaves from the hybrid ADA313 grown up to the four-leaf stage after low-temperature treatment were subjected to chip hybridization with 321 known maize miRNAs. The results indicated that miR408 and miR528 are low-temperature response-related miRNAs and that target genes of miR319 and miR396 may be involved in leaf development under low-temperature stress [61]. Furthermore, overexpression of more than 700 genes in maize revealed that *ZmRR1*, a type-A response regulator of the cytokinin signaling pathway, significantly increased cold tolerance during seeding in transgenic maize, whereas its mutation led to cold sensitivity. Moreover, candidate gene association analysis revealed a 45 bp insertion/deletion (InDel-35) in the coding region of *ZmRR1* is significantly associated with natural variation in cold tolerance in maize [66].

## 4. Trends in Maize Chilling Tolerance-Related Research

Using “chilling tolerance AND maize” as the keywords, we searched for NCBI literature and obtained nine research papers published before 2000. Among them, there were six papers investigating parameters of the antioxidant system as indicators after low-temperature treatment, one paper investigating the photosynthetic physiology for indicators and three, two and zero papers investigating the membrane system, hormone-level changes and osmoregulatory substances as indicators of chilling tolerance, respectively. Three papers investigated more than two indicators (e.g., both membrane system and antioxidant system indicators). Nine maize cold-tolerance-related research papers were published between 2001 and 2012, among which four investigated the antioxidant system, five investigated the photosynthetic physiology and four, two and two investigated parameters related to the membrane system, hormone level changes and osmoregulatory substances, respectively. Furthermore, four papers investigated more than two indicators.

In terms of the most recent research, 41 papers were published in 2013–2022, including 13 and 15 papers investigating the antioxidant system and photosynthetic physiology, respectively, while five, five and eight papers investigated membrane system-related parameters, hormone level changes and osmoregulatory substances. Twelve articles investigated more than two indicators involved in cold tolerance. Among them, 11 papers used populations rather than a few inbred lines or hybrids as the experimental material for cold-tolerance gene mining. Therefore, in the last 10 years (2013–2022), the number of research papers published on cold tolerance in maize has been greater than the sum of papers published in the previous 30 years. Moreover, the percentage of papers related to indicators of the membrane system, antioxidant system and photosynthetic system parameters was the highest across all periods, indicating that a low temperature can significantly affect maize through these three indicators. The number of papers with photosynthetic parameters and osmoregulatory substances as cold tolerance indicators demonstrated an increasing trend year by year. Meanwhile, the proportion of papers combining multiple indicators (≥2 indicators) was roughly the same in all periods (Figure 2). The cold treatment temperature ranged from 2 to20 °C (Table 1 and Table 2).

In terms of the research articles’ content, studies before 2005 mostly focused on measuring physiological and biochemical indicators before and after low-temperature treatment using a single genotype. Further studies compared the differences in relevant parameters between low-temperature-tolerant and -sensitive maize before and after low-temperature treatment. In cold-tolerant genetic materials, antioxidant-related enzymes (such as superoxide dismutase, peroxidase, catalase and glutathione reductase) exhibited higher activity and caused less damage to the photosynthetic system. The osmoregulatory compound content is relatively higher after low-temperature treatment. Although the number of maize varieties studied has been limited (most studies focused on only one or a few inbred lines/hybrids), these parameters reflect the cold tolerance of maize to a certain extent. In addition, measures can be taken to improve maize cold tolerance based on the changes in these physiological and biochemical indicators. For example, regulating the compounds of unsaturated fatty acids can reduce electrolyte leakage and preserve cells’ membrane structure under low-temperature stress. Treatment of maize seeds or seedlings with melatonin, chitosan, salicylic acid and ABA has been shown to maintain the dynamic balance of the antioxidant system under low-temperature stress. It reduces photoinhibition and increases osmoregulatory substances such as photosynthetic products (soluble sugars) to enhance the cold tolerance (Table 1).

All these “exogenous” applications have the potential to enhance the cold tolerance of maize. However, genetic improvement can improve cold tolerance in maize at its core more effectively and permanently. The alterations in physiological and biochemical parameters before and after the low-temperature treatment can serve as the phenotypic basis for exploring the underlying genetic loci controlling chilling tolerance. Based on phenotypic differences, recombinant inbred populations, F_2:3_ populations and diploid populations have been generated from cold-tolerant and -sensitive maize lines. Linkage analyses were performed to identify QTLs that contribute significantly to cold tolerance in maize using several hundred to thousands of SSR (simple sequence repeat), RFLP (restriction length polymorphism fragments) or SNP markers. Due to the low marker density and the large confidence intervals of QTLs (in the range of several megabasesor even larger), the studies were limited to only QTL identification. Papers published after 2020 introduced RNA-seq data of biparental samples before and after low-temperature treatment into the linkage analysis to locate cold-tolerance-related QTLs more precisely. In addition, the sequence information (including nucleotide sequence and amino acid sequence) of genes contained within the candidate QTL, their expression levels before and after low-temperature treatment and functional annotation information of homologous genes were comprehensively analyzed. Consequently, several to several-dozen genes induced by low-temperature stress and exhibiting significant sequence and expression level differences in biparental samples were put forward as maize cold-tolerance-related candidate genes (Table 2).

The advancement of high-throughput sequencing technology and the completion of sequencing assembly and annotation of the maize inbred line B73 allowed researchers to focus on the global maize germplasm, for which they began to collect and analyze large samples and data. The association analysis approach, originally developed and used in human disease research, is increasingly applied to plant genetic studies [67]. Its advantages in resolving quantitative traits are becoming increasingly evident. Cold-tolerance traits are complex quantitative traits, and association analysis was implemented late in this field. The first report of genome-wide association analysis in maize cold-tolerance-related research was published in 2013 [55]. Afterward, studies were performed using the association analysis method to determine maize cold-tolerance-related genes during germination and seedling stages. The numbers of SNPs used for genome mapping rose to the “ten thousand” and “million” levels. The high density of molecular markers allows for the direct tracing of significant SNP loci to specific genes. Due to the short range of linkage disequilibrium decay in the maize genome, the QTL interval range where significant SNP loci are located is comparatively small and can usually be narrowed down to a few hundred Kb. After 2016, various combinations (association analysis with linkage analysis, association analysis with RNA-seq and genome-wide linkage analysis with candidate gene association analysis) emerged to further narrow the candidate genes identified by association analysis (Table 2). However, these studies usually stopped at the cold-tolerance gene-association stage, and few putative candidate genes have been functionally validated. In 2021, a landmark study investigated the involvement of the *ZmRR1* gene in maize cold tolerance and provided an in-depth analysis of its molecular mechanism. The mitogen-activated protein kinase ZmMPK8 phosphorylates the serine locus of ZmRR1, thus promoting the degradation of ZmRR. A 45-bp deletion in the coding region of ZmRR1 contains the locus phosphorylated by ZmMPK8, thus inhibiting the degradation of ZmRR1 and enabling maize to achieve greater cold tolerance. Analysis of transcriptome data revealed that *ZmRR1* could upregulate the expression of *ZmDREB1* and *CesA* (*cellulose synthase A*) to improve maize cold tolerance. This is the only work to date on the underlying molecular mechanisms of natural variation in maize cold-tolerance-related genes expressed at the seedling stage, pushing maize cold-tolerance-related research to a newlevel.

## 5. Summary and Outlook

Temperature is a key limiting factor for plant growth and reproduction under natural environmental conditions. Maize was originally planted in tropical and subtropical regions due to its optimal germination and growth under warm conditions. Its successful introduction in temperate regions was achieved by shortening its growing period. As a result, it has not been fully adapted to temperate regions. It often suffers from low-temperature stress such as late-spring cold spells and frosts, which can seriously affect the yield and quality of maize in the later stages and threaten food security.

International breeding programs to improve maize cold to lerance were started as early as 1914. Yet, it was not until 1949 that Neal first reported that different genotypes of maize exhibited significant variation in low-temperature tolerance at the germination stage [68]. Researchers initially focused their research on the changes in antioxidant enzymes, photosynthetic physiology-related parameters and the membrane system before and after low-temperature treatment. Using inbred lines and hybrids with different low-temperature tolerances, the physiological and biochemical indicators related to cold tolerance were identified. In 2005, linkage analysis was used in an F_2:3_ population constructed from parents with significant differences in low-temperature tolerance to identify QTLs associated with cold tolerance. This marked the beginning of efforts to explore and reveal the molecular genetic basis for the cold-tolerance phenotype in maize. Subsequently, association analyses could directly link loci significantly associated with cold tolerance to candidate genes due to the advancement of high-throughput sequencing, the reduction of sequencing costs and the proliferation of marker numbers and densities. Combining RNA-seq data further narrowed down the candidate genes (Table 2). However, these putative candidate genes have not been functionally characterized. we are still yet to see a comprehensive dissection of the maize chilling-tolerance-related signaling pathways.

Elucidation of the cold signaling pathways was initially based on studies on the model plant *A**. thaliana*. In the early stage of low-temperature stress signaling, calcium channels and the plasma membrane protein CRLK1/2 (CaM-regulated receptor-like kinase) are activated, triggering the rapid transmission of calcium signals and the onset of the MAPK cascade response. Calcium signals are decoded by a series of calcium-binding proteins, including CaM (calmodulin), CML (CaM-like protein), CDPKs (Ca^2+^-dependent protein kinases) and CBLs (calcineurin B-like proteins), triggering the *inducer of CBF expression-C repeat binding factor-cold regulated* (*ICE1-CBF-COR*) in response to low-temperature stress. This core cold signal transduction pathway in *Arabidopsis* has been intensively studied at the transcriptional and post-translational levels. ICE1 was degraded by ubiquitination of HOS1 (high expression of osmotically responsive gene 1). On the other hand, sumoylation of ICE1 by SIZ1 (SAP and Miz 1) and phosphorylation of ICE1 by OST1 (open stomata 1) inhibit the degradation of ICE1. OST1 phosphorylates two other proteins, BTF3 (basic transcription factor 3) and BTF3L (BTF3-like protein), enabling them to interact with CBF, thus enhancing the stability of CBF under low-temperature stress. During cold acclimation, ICE1 is phosphorylated and degraded by MAPK3/6. A 14-3-3 protein is phosphorylated by CRPK1 (cold-responsive protein kinase 1), which shuttles from the cytoplasm to the nucleus and promotes the degradation of CBF to regulate the cold response process (reviewed by [69,70]). CBF binds to the conserved CRT/DRE (C-repeat/dehydration response element) element in the promoter region of the *COR* gene and promotes its expression.

A landmark study of the cold-signaling pathway was that of the COLD1 (chilling tolerance divergence 1) protein in rice [71]. Low-temperature stimulation triggers a conformational change in COLD1; COLD1 leads to the flow of extracellular calcium ions into the cell, ROS production, ABA accumulation and the MAPK cascade reaction via COLD1-RGA1 interaction. Furthermore, calcium signals are delivered by calcium-binding proteins, which upregulate the transcription factors *OsMYB3R-2*, *OsMYBS3* and *OsMADS57*. Under low-temperature conditions, OsMAPK3 phosphorylates OsbHLH002/OsICE1, inhibiting the degradation of OsICE1 (opposite MAPK3/6 phosphorylation of ICE1 in *A**. thaliana*, which promotes ICE1 degradation), and also promotes the expression of OsTPP1, leading to an increase in the level of trehalose, thereby enhancing cold tolerance in rice. Therefore, the *OsMAPK3*–*OsbHLH002*/*OsICE1*–*OsTPP1* pathway is a crucial low-temperature response pathway in rice (reviewed by [72,73]).

Compared with research in *A**. thaliana* and rice, maize-related cold tolerance research is still in its infancy. The mining and identification of chilling tolerance genes is “sporadic” rather than “systematic”. Most candidate genes are still at the “prediction” stage, with very few genes functionally validated. So far, only four genes, *ZmRR1*, *ZmPgb1.1*, *ZmASR1* and *RAF1*–*LSSS*, have been validated in transgenic maize. *ZmMYB31* and *ZmMYB–**IF35* have been functionally characterized in *A**. thaliana* and *ZmMKK1* in tobacco, conferring higher cold tolerance in transgenic plants. These findings have provided insights into research on cold tolerance in maize (Figure 3). The dramatic increase in the number of research papers related to chilling tolerance in maize published in the last 10 years compared to the previous trend also indicates that research in this field is intensifying.

Under chilling stress, overexpression of *ZmMKK1* or *ZmPgb1.1* enhances the activity of antioxidant enzymes such as sodium dismutase and catalase to alleviate ROS-induced damage. In addition, *ZmMKK1* suppresses the increase of malondialdehyde content and relative electrolyte leakage to protect biological membrane stability while increasing osmoregulatory substances such as proline and soluble sugar. Overexpression of *ZmPgb1.1* inhibits BR biosynthesis. Meanwhile, overexpression of *ZmMYB–**IF35*, *ZmASR1* or *ZmMYB31* reduces photoinhibition by a slow decrease in Fv/Fm and an increase in sodium dismutase activity. Alternatively, overexpression of *ZmMYB*–*IF35* or *ZmRR1* results in low ion leakage levels, whichmitigate membrane damage caused by low temperature. In transgenic lines overexpressing *RAF1*–*LSSS*, photosynthesis parameters such as Φ_psII_, Φ_II_ and Fv/Fm slightly decrease to maintain a low degree of photoinhibition. In Figure 3, arrows indicate increases and T-shapes indicate decreases. Solid arrows and T-shapes represent the involvement of multiple genes, and hollow arrows and T-shapes represent the involvement of individual genes.

Future research should be expanded in the following aspects: (1) more emphasis should be givento the validation of maize chilling-tolerance-related genes, to analyze the functions of key molecular modules consisting of important regulatory factors and their upstream and downstream counterparts in the cold-tolerance response. This will elucidate the entire transmission chain of low-temperature signals from perception to transmission, to comprehensively analyze maize chilling signaling pathways. The cross-talk between the low-temperature signaling and other signaling pathways such as drought, salt and plant hormone pathways should be assessed to enrich our knowledge of the complex regulatory networks underlying the mechanisms of chilling responses; (2) focus on sequencing a large number of maize inbred lines, including their ancestor teosinte, to enrich natural variation sequence information. This will allow us to dig deeper into the favorable alleles of cold-tolerance-related genes, analyze the genetic evolutionary mechanism of cold tolerance in maize and decipher the molecular mechanisms underlying heterosis in maize chilling tolerance; (3) maize genetic transformation technologies require further breakthroughs as they are strongly affected by the genotype. Genomic selection-based breeding, CRISPR and other methods should be combined to aggregate favorable alleles of cold-tolerance genes and their signaling modules into economically important maize varieties. This could accelerate the utilization of cold-tolerance genes, enable the breeding of a new maize cold-tolerance germplasm and lay the foundation for maintaining high and stable yields even after exposure to chilling stress conditions.

## Figures and Tables

**Figure 1 plants-11-02082-f001:**
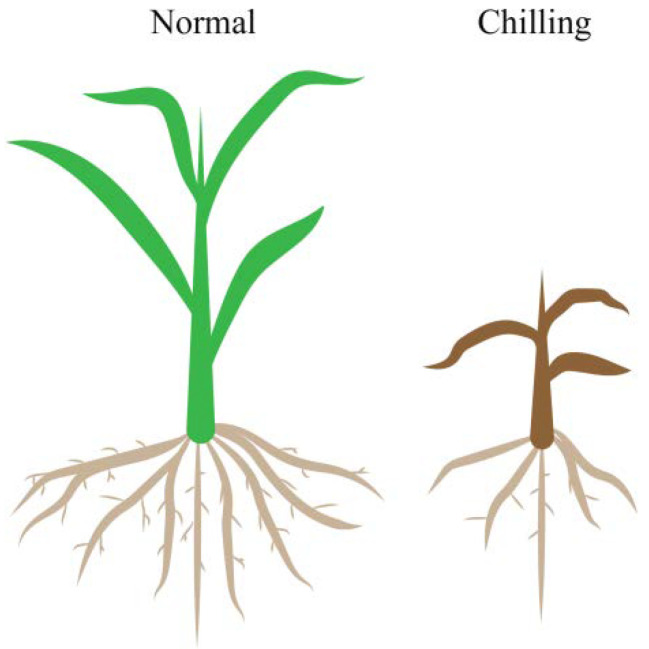
Schematic diagram of maize growth under normal and chilled conditions.

**Figure 2 plants-11-02082-f002:**
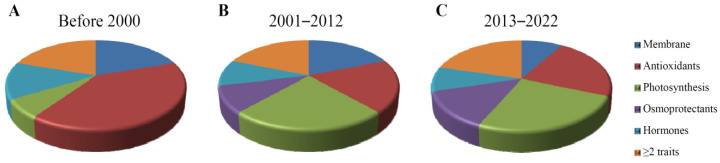
Number of peer-reviewed articles published on maize chilling tolerance before 2000 (**A**), during 2001–2012 (**B**) and during 2013–2022 (**C**).

**Figure 3 plants-11-02082-f003:**
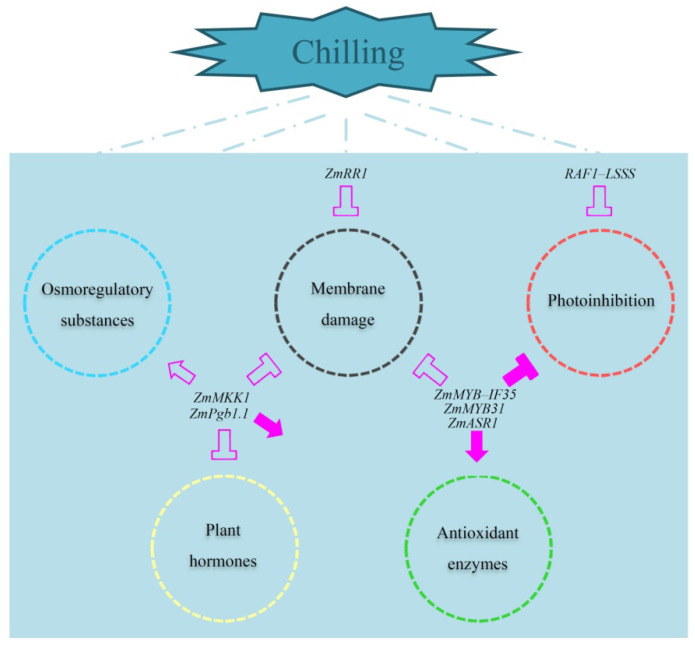
Published genes involved in the changes in terms of osmoregulatory substances, membrane damage, photoinhibition, plant hormones and antioxidant enzymes under chilling stress.

**Table 1 plants-11-02082-t001:** Measurement of physiological and biochemical indices under low-temperature treatment.

Year	Low-Temperature Treated		Indices					Ref.
	Material (Germplasm)	Condition	Membrane	Antioxidants	Photosynthesis	Osmoprotectants	Hormones	
1990	10-day-old seedlings (*Zea mays* L. cv. LG11)	5 °C for 6 h			Fv/Fm			[33]
1993	2-week-old seedlings (four inbred lines: Z 7, Mo 17, KW1074 and Penjalinan)	14/12 °C day/night for 4 days and then chilled for 5 days at 5/3 °C	ion efflux				abscisic acid	[14]
1993	2- to 3-week-old seedlings (F7-Rp III )	none	free fatty acids, oxygen evolution					[17]
1993	maize suspension-cultured cells(*Zea mays* L. cv Black Mexican Sweet)	4 °C for 12 h, 24 h, 2 days and 4 days					abscisic acid	[46]
1994	3-day-old seedlings (Pioneer inbred G50)	14 °C for 3 days, then 4 °C for 7 days		catalase 3,hydrogen peroxide				[19]
1994	3-day-old seedlings (Pioneer inbred G50)	14 °C for 3 days, then 4 °C for 7 days, or just 4 °C for 7 days		catalase 3, peroxidase			abscisic acid	[47]
1997	11-day-old seedlings (chilling-susceptible CO 316 and chilling-tolerant CO 328 inbred lines)	6/2 °C day/night for 24 and 48 h		superoxide dismutase, glutathione reductase, ascorbate peroxidase				[8]
1997	3-day-old seedlings (Pioneer inbred G50)	14 °C for 1 day or 4 °C for 1 day followed by recovery at 27 °C for 1 day, then 4 °C for 7 days		catalase				[27]
1999	seed to 21- to 23-day-old seedlings (three F2 from the cross B73×IABO78, as well as the two parental lines, chilling-sensitive germination B73 and chilling-tolerant germination IABO78)	14 °C	mitochondrial inner membranes, 18-carbon unsaturated fatty acids, fluidity	cytochrome c oxidase mitochondrial peroxidase				[13]
2000	3-leaf-stage seedlings (*Zea mays* L. cv. H99)	seed was germinated for 5 days, then transferred at 20, 18, 15 or 10 °C until the third leaf was expanded		oxidative damage				[31]
2002	2-leaf-stage seedlings (chilling-sensitive Penjalinan and chilling-tolerant Z7 inbred lines)	5 °C for 7 days		glutathione, glutathione reductase				[20]
2002	2-week-old seedlings (*Zea mays* L. cv. Golden Jubilee)	2.5 °C for 1–4 days	electrolyte leakage	glutathione reductase, guaiacol peroxidase			salicylic acid	[49]
2004	10-to 15-day-old seedlings (variety of *Zea mays* L.)	4 °C for 1 to 5 h, sampled at 1-hr intervals			PSII activity			[32]
2005	root cortex protoplasts (chilling-sensitive Penjalinan and chilling-tolerant Z7 inbred lines)	5 °C for 3 days	hydraulic conductance	hydrogen peroxide				[11]
2005	seed to 3-leaf-stage seedlings (chilling-tolerant KW 1074 and chilling-sensitive CM 109 inbred lines)	14/12 °C day/night			photosynthetic efficiency			[36]
2009	seed (chilling-tolerant HuangC and chilling-sensitive Mo17 inbred lines)	5 °C for 3 days	relative permeability of the plasma membrane	peroxidase, catalase, malondialdehyde		soluble sugars, proline		[23]
2010	3-leaf seedlings (chilling-tolerant KW 1074 and chilling-sensitive CM 109 inbred lines)	14/12 °C day/night for either 4 or 28 h			photosynthesis			[37]
2012	10-day-old seedlings (LM-17 inbred lines)	maximum and minimum temperature in net house ranged between 17.6 and 24.5 °C and 2.8 and 7.4 °C; treated for 7, 14 and 21 days	tissue water content, membrane injury index		total chlorophyll	soluble sugar, protein content	24-epibrassinoslide	[52]
2014	8-week-old transgenic tobacco	12 °C for 2 and 4 days		antioxidant enzyme activities, ROS-related genes		osmoregulatory substances		[30]
2015	seed (chilling-tolerant HuangC and chilling-sensitive Mo17 inbred lines)	5 °C for 3 days		protective enzyme activities, malondialdehyde content				[24]
2016	third-leaf seedlings (three inbred lines: S68911, S50676 and S160)	14/12 °C day/night for 4 days, followed by 4 days at 8/6 °C day/night			photosynthetic apparatus			[39]
2016	5-leaf-stage seedlings (three flint lines: F2, F283, F03802; three dent lines: F353, B73, Mo17; two hybrids: F03802xF353, F2xF353)	10/7 °C day/night (inbred lines) or 10/4 °C day/night (hybrids) for one week			chlorophyll biosynthesis, CO_2_ assimilation			[34]
2017	seed (hybrids A and B provided by Limagrain Europe)	5, 10, 15 and 18 °C for germination assay; 10 and 18 °C for 24 h for electrolyte leakage measurements, total lipid extraction and phospholipid analysis	saturated or poorly unsaturated fatty acids, electrolyte leakage					[12]
2017	2-week-old seedlings (He 344)	5 °C for 3 days	membrane lipid adjustment					[15]
2017	3-leaf-stage seedlings (chilling-tolerant KW 1074 and chilling-sensitive CM 109)	14/12 °C for either 1, 4, 28, or 168 h				sucrose		[41]
2017	six-leaf seedlings (hybrid Dekalb-6789)	average chilling temperature was 13–8 °C from sowing to harvesting	electrolyte leakage					[50]
2017	seed (Meiyuno.3)	13 °C for 7 days		reactive oxygen species			abscisic acid	[51]
2018	3-leaf-stage seedlings (chilling-tolerant S68911 and chilling-sensitive B73)	12–14 °C day/night for 28 h and 3 days			net CO_2_ assimilation, F’v/F’m, Fv/Fm, Φ_PSII_			[18]
2018	2-week-old seedlings (*Zea mays* L. cv. Colisee)	12–14 °C for 2 weeks		superoxide dismutase activity, antioxidants, H_2_O_2_		proline		[25]
2018	3-week-old seedlings (*Zea mays* L. cv. Jidan 198 and Jinyu 5)	5 °C for 2 days		superoxide dismutase, peroxidase activity, malondialdehyde content	Fv/Fm			[28]
2018	13-day-old-seedlings (*Zea mays* L. hybrid Norma)	15/13 °C day/night for 3 days under three different light conditions, then 5 °C for 3 days, then back to 22/20 °C day/night for a 1-day recovery period			photoinhibition	soluble sugars		[35]
2019	4-leaf-stage seedlings (chilling-tolerant M54 and chilling-sensitive753F inbred lines)	4 °C for 0, 4 and 24 h	unsaturated fatty acid		PSII, secondary metabolites			[16]
2019	seed (*Zea mays* L. cv. Jinongnuo 112)	13 °C for 12 and 24 h		hydrogen peroxide, superoxide dismutase, peroxidase, catalase, ascorbate peroxidase, malondialdehyde concentrations				[22]
2019	2-week-old transgenic *Arabidopsis*	4 °C for 3, 6, 9 and 12 h	ion leakage	antioxidant enzyme activity, reactive oxygen species	Fv/Fm			[1]
2019	2-week-old transgenic *Arabidopsis*	4 °C for 3, 6, 9 and 12 h		superoxide dismutase, ascorbate peroxidase, reactive oxygen species	Fv/Fm			[29]
2020	3-leaf-stage seedlings (B73 and W22 background)	10/8 °C day/night for 4 days				raffinose biosynthesis		[44]
2020	seeds (three chilling-tolerant germination lines: 91, 64, 63, and three chilling-sensitive germination lines: 44, 54, 57), as well as their hybrid combination by reciprocal crosses	10 °C for 4 and 7 days		catalase, esterase enzymes				[21]
2020	3-leaf-stage seedlings (Q319 and DA-6 inbred lines)	11±1 °C for 0, 1, 3, 5 and 7 days		oxygen metabolism	photosynthesis			[26]
2020	11-day-old seedlings (two maize hybrids; Xida889 and Xida319, and two maize inbred; Yu13 and Yu37)	15/12 °C day/night for 12 days		total antioxidant capability, superoxide dismutase, peroxidase, catalase and glutathione reductase activities				[5]
2020	3-week-old seedlings (transgenic maize)	14 °C/12 °C day/night for 2 weeks			photochemical quenching			[40]
2020	4-day-old seedlings (chilling-tolerant CFD04_349 and chilling-sensitive CFD04_332; two double-haploid (DH) population derived from the F1 cross between F353 and D09)	15 °C/11 °C day/night for about 8 weeks			chlorophyll content, glucose-6-phosphate dehydrogenase activity	sucrose-to-starch ratio		[43]
2020	11-day-old seedlings	15/13 °C day/night for 3 days, followed by 5 °C for 3 days					salicylic acid	[45]
2021	2-week-old seedlings (chilling-tolerant Gurez local and chilling-sensitive Gujarat-Maize-6)	6 °C for 2, 4, 6, 8, 10 and 12 h		hydrogen peroxide, malondialdehyde		free proline, total protein, total soluble sugars, trehalose, total phenolics, glycine betaine		[42]
2021	3-leaf-stage seedlings (B73, B104, CM7, CM37, CML77, CML333, M37W, Mt42, NC300, R177, and Tzi9 inbred lines)	4 °C for 3–7 days					abscisic acid	[48]
2021	2-leaf-stage seedlings (transgenic maize)	10 °C/4 °C day/night for 72 h					brassinosteroid biosynthetic and response genes	[53]
2022	seed (chilling-tolerant 04Qun0522-1-1 and chilling-sensitive B283-1 inbred lines)	13 °C for 4 days		antioxidant metabolism-related pathways	photosynthetic system			[38]

**Table 2 plants-11-02082-t002:** Identification of QTLs/genes underlying chillingtolerance.

Year	Low-Temperature Treated		Type of Marker	Number of Markers	Trait	Method	Number of QTL/Loci	Candidate Genes	Ref.
	Material (Germplasm)	Condition							
2005	from seed to 2-leaf-stage seedlings (226 F2:3 population, chilling-tolerant ETH-DH7 and chilling-sensitive ETH-DL3 as parents)	field experiments; plants sown early were exposed to low temperature	SSR		Φ_PSII_, Fv/Fm, SPAD, Fo, Fm, SDW, LAT, N%, C%, C:N	linkage analysis	6, 5, 4, 1, 2, 4, 2, 2 and 3 QTLs related to SPAD, Φ_PSII_, Fv/Fm, Fo, Fm, LAT, SDW, N% and C:N, respectively	none	[54]
2013	from seed to 3-leaf-stage seedlings (375 maize inbreeds)	growth chamber (16/13 °C, day/night) and field experiments(mean temperature after sowing for 30 days about 15 °C)	SNP	56,110	Φ_PSII_, Fv/Fm, SPAD, SDW, LDW, LAS, SLA, RGR	association analysis	16 QTLs	24 candidate genes: GRMZM2G371167, GRMZM2G049609, GRMZM2G151811, GRMZM2G059165, GRMZM2G103773, GRMZM2G103843, GRMZM2G035584, GRMZM2G099850, GRMZM2G123790, GRMZM2G328742, GRMZM2G394827, GRMZM2G094892, GRMZM2G093346, GRMZM2G381059, GRMZM2G167856, GRMZM2G050649, GRMZM2G130442, GRMZM2G349709, GRMZM2G057386, GRMZM2G057231, GRMZM2G358161, GRMZM2G057709, GRMZM2G021388 and GRMZM2G021277	[55]
2016	from seed to 8-day-old seedlings (208 and 212 F_10_ RILs derived from two crosses, Yu82×Shen137 and Yu537A×Shen137	18 ± 1 °C	SNP	1172 SNPs for RIL from Yu82×Shen137 and 1139 SNPs for Yu537A×Shen137parents, respectively	GP, GI, SL, SVI, MGT	linkage analysis	5 mQTLs	none	[56]
2016	from seeds to 2-leaf-stage seedlings (two panels of 306 dent and 292 flint maize inbred lines)	14/8 °C day/night	SNP	49,585	DTE, SPAD in the second leaf,Φ_PSII_, EV	association analysis	9 QTLs	36 candidate genes: GRMZM2G061206, GRMZM2G061127, GRMZM2G174274, GRMZM2G174249, GRMZM2G174221, GRMZM2G174196, GRMZM2G174137, GRMZM2G074241, GRMZM2G375807, GRMZM2G419024, GRMZM5G899800, GRMZM2G416069, GRMZM2G115730, GRMZM2G115750, GRMZM2G130043, GRMZM2G130002, GRMZM2G129979, GRMZM2G178398, GRMZM2G172244, GRMZM2G171420, GRMZM2G171394, GRMZM2G078143, GRMZM2G084825, GRMZM2G154216, GRMZM2G341036, GRMZM2G102862, GRMZM2G405090, GRMZM2G127510, GRMZM2G127499, GRMZM2G429396, GRMZM2G124794, GRMZM2G423478, GRMZM2G180027, GRMZM2G480480, GRMZM2G180080 and GRMZM2G180082	[57]
2017	seed (282 inbred lines)	8 °C (chilling) and 25 °C (normal)	SNP	2 × 10^6^	GR, DT50, GI, RGR, RDT50, RGI	association analysis	17 genetic loci	18 candidate genes: GRMZM2G704005, GRMZM2G113158, GRMZM2G318156, GRMZM2G012148, GRMZM2G300994, GRMZM5G871707, GRMZM2G462797, GRMZM2G178486, GRMZM5G806387, GRMZM2G148793, GRMZM2G389768, GRMZM2G073535, GRMZM5G802338, GRMZM2G057186, GRMZM2G081928, GRMZM2G019746, GRMZM2G033884 and GRMZM2G170890	[3]
2017	root of post-germination (four cultivars: chilling-tolerant Picker and PR39B29, chilling-sensitive Fergus and Codisco)	germination for 1, 2, 3, 4 and5 days	oligo array	46,000	growth ratio	microarray analysis		*MZ00004486*	[58]
2017	3-leaf-stage seedlings (338 testcrosses and an F_2:3_ population)	5.5–6.5 °C for 7 days	SNP and SSR	556,809 SNP markers for association analysis, 152 SSR markers for linkage map construction	LRD, WCS, RRS, SSC	association analysis and linkage analysis	32 significant loci for association analysis,7 QTL for linkage analysis	36 stress tolerance-related candidate genes: GRMZM2G460383, GRMZM2G363229, GRMZM2G082097, GRMZM2G032209, GRMZM2G110242, GRMZM2G159756, GRMZM2G470984, GRMZM2G053384, GRMZM2G000936, GRMZM2G035807, GRMZM2G102927, GRMZM2G102811, GRMZM2G457267, GRMZM2G332258, GRMZM2G110085, GRMZM2G058518, GRMZM2G437460, GRMZM2G580389, GRMZM2G463462, GRMZM2G403609, GRMZM2G132882, GRMZM2G111696, GRMZM2G411288, GRMZM2G019986, GRMZM2G407825, GRMZM2G107481, GRMZM2G051917, GRMZM2G053206, GRMZM2G092327, GRMZM2G138161, GRMZM2G348512, GRMZM2G121878, GRMZM2G027098, GRMZM2G012479, GRMZM2G000404 and GRMZM2G395535	[59]
2020	seed (222 diverse inbred lines)	10 °C for 31 days, at 3-day intervals	SNP	40,697	RGR, RGL, RRL, RRSA, RRV, RGI, RVI, RSVI, XYRGR, XYRGL, XYRSVI, KSRGR, KSRGL, KSRSVI	association analysis and RNA-seq	30 significant SNPs	82 candidate genes associated with significant SNPs; Zm00001d039219 and Zm00001d034319 were further identified by RNA-seq	[60]
2020	4-leaf-stage seedlings (*Zea mays* L. hybrid ADA313)	25/4 °C day/night for 2 days	miRNAs	321	LL, LA4, LER, Clma, CLme, Length of the cell at the end of meristem, *P*, *D*, *Tc*, *Lmer*, *Lgr*, *Lel*, *Nmer*, *Ngz*, *Nel*, *Tel*, *Rel*	microarray analysis		miR408, miR528 and target genes of miR319 and miR396	[61]
2021	seed (300 inbred lines)	10 °C for 10 days	SNP	43,943	FG, TG, RL, SL, RRS	association analysis, candidate gene association study and expression pattern analysis	15 significant SNPs	Zm00001d010454, Zm00001d010458, Zm00001d010459 and Zm00001d050021	[62]
2021	from seed to 2-leaf-stage seedlings (836 maize inbreeds)	14/10 °C day/night	SNP	156,164	DTE, DTSL, EV, SPAD, Fv/Fm, DW	association analysis	159 QTLs	226 candidate genes	[63]
2021	14-day-old seedlings (97 RILs of IBM Syn4 derived from B73 × Mo17)	4 °C for 8 h	RFLP and SSR	over 1850	chlorophyll concentration, leaf color, tissue damage	linkage analysis and RNA-seq	2 QTLs	27 candidate genes: GRMZM2G003506, GRMZM2G331652, GRMZM2G331638, GRMZM2G155242, GRMZM2G107774, GRMZM2G103079, GRMZM2G395121, GRMZM2G094444, GRMZM2G098714, GRMZM2G173067, GRMZM2G163043, GRMZM2G095382, GRMZM2G014560, GRMZM2G096753, GRMZM2G170692, GRMZM2G153488, GRMZM2G153359, GRMZM2G153263, GRMZM2G098474, GRMZM2G165290, GRMZM2G175177, GRMZM2G159904, GRMZM5G841914, GRMZM2G178603, GRMZM2G178509, GRMZM2G178497 and GRMZM2G139837	[64]
2022	seed (176 B73 × Mo17 (IBM) Syn10 doubled haploid (DH) population)	10 °C for 21 days	SNP	6618	LTPL, LTSL, LTRL, LTGR, LTGI, LTVI, LTSVI, LTAGD	linkage analysis and RNA-seq	7 QTLs	6 candidate genes: Zm00001d043166, Zm00001d007315, Zm00001d027974, Zm00001d027976, Zm00001d007311 and Zm00001d053703	[65]

## Data Availability

Not applicable.

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
