# Peer review of "Chilling Tolerance in Maize: Insights into Advances—Toward Physio-Biochemical Responses’ and QTL/Genes’ Identification"

_plants, 2022, doi:10.3390/plants11162082_

Round 1
Reviewer 1 Report
The authors of this manuscript discuss developments in improving Maize's chilling tolerance. In JXB, a similar article was published (https://doi.org/10.1093/jxb/erac045). Furthermore, the authors discussed progress made in identifying cold-tolerance QTLs (quantitative trait loci) and genes for genetically improving maize chilling tolerance. Maize chilling tolerance research was highlighted from 1990 to 2022 by the authors. The other parts of the review like effect of chilling stress on osmoregulatory, plant hormone levels in maize were also included. With a few minor revision, the review can be accepted:
1. Introduction can be extended.
2. Authors can add a Figure showing the effect of chilling stress on the phenotype of the maize plants.
3. Also author can also include studies based on effect of chilling stress on roots.
Reviewer 2 Report
The manuscript entitles “Chilling Tolerance in Maize: Insights into Advances -towards Physio-Biochemical Responses and QTL/Genes Identification” has been written good and have some comments below:
In abstract, Line no 12: The sentence should be start like, “This review covered the latest research and progress………………….”
Line no 13: Remove word “We” The sentence should not be start like, We found etc., It should be like The present review focused…… Also remove “We” words from throughout the paper.
Keywords should be different from the title.
Introduction part is very short, and Please provide the reference for the sentence starts from Line no 33…
Line no 124 and 125, Arabidopsis thaliana, should be A. thaliana.
Line no 149. Arabidopsis thaliana should be A. thaliana .
In table no 1, Please remove the reference no 27 given two times in table.
In table 2, The abbreviation forms should be below the table horizontally.
In figure 2, legends should be correctly mention.
In References, line no 620 and 622. Evidence for Chilling-Induced Oxidative Stress in Maize Seedlings and a Regulatory Role for Hydrogen Peroxide. It should be in small letters.
References should be according to the journals guideline. And follow same format for all the references.
English language must be improved in throughout the manuscript.
